# Measuring geographical disparities in England at the time of COVID-19: results using a composite indicator of population vulnerability

Catia Nicodemo [1,2] Samira Barzin,[3,4] Nicolo' Cavalli,[5,6] Daniel Lasserson [7] Francesco Moscone,[8,9] Stuart Redding,[1,2] Mujaheed Shaikh[10]

For numbered affiliations see end of article.

**Correspondence to**
Dr Catia Nicodemo;
catia.nicodemo@economics.ox.ac.uk

## ABSTRACT

**Objectives** The growth of COVID-19 infections in England raises questions about system vulnerability. Several factors that vary across geographies, such as age, existing disease prevalence, medical resource availability and deprivation, can trigger adverse effects on the National Health System during a pandemic. In this paper, we present data on these factors and combine them to create an index to show which areas are more exposed. This technique can help policy makers to moderate the impact of similar pandemics.

**Design** We combine several sources of data, which describe specific risk factors linked with the outbreak of a respiratory pathogen, that could leave local areas vulnerable to the harmful consequences of large-scale outbreaks of contagious diseases. We combine these measures to generate an index of community-level vulnerability.

**Setting** 91 Clinical Commissioning Groups (CCGs) in England.

**Main outcome measures** We merge 15 measures spatially to generate an index of community-level vulnerability. These measures cover prevalence rates of high-risk diseases; proxies for the at-risk population density; availability of staff and quality of healthcare facilities.

**Results** We find that 80% of CCGs that score in the highest quartile of vulnerability are located in the North of England (24 out of 30). Here, vulnerability stems from a faster rate of population ageing and from the widespread presence of underlying at-risk diseases. These same areas, especially the North-East Coast areas of Lancashire, also appear vulnerable to adverse shocks to healthcare supply due to tighter labour markets for healthcare personnel. Importantly, our index correlates with a measure of social deprivation, indicating that these communities suffer from long-standing lack of economic opportunities and are characterised by low public and private resource endowments.

**Conclusions** Evidence-based policy is crucial to mitigate the health impact of pandemics such as COVID-19. While current attention focuses on curbing rates of contagion, we introduce a vulnerability index combining data that can help policy makers identify the most vulnerable communities. We find that this index is positively correlated with COVID-19 deaths and it can thus be used to guide targeted capacity building. These results suggest that a stronger focus on deprived and vulnerable communities is needed to tackle future threats from emerging and re-emerging infectious disease.

### Strengths and limitations of this study

► We provide important information to help identify the communities most vulnerable to harmful effects of COVID-19.
► This fills an important gap in the literature, with only a handful of previous studies that show the distribution of underlying risks within national systems.
► Judgement was required when deciding which variables to include or omit.
► Our methods give equal weight to each variable when creating the index.
► Clinical Commissioning Groups are quite large and using smaller areas may be more appropriate.

## INTRODUCTION

The current COVID-19 outbreak is triggering a renewed understanding of health risks and underlying health vulnerabilities.[1] In a pandemic, overlooked vulnerabilities may arise from the social and biological makeup of local communities. Accumulated evidence from emerging and re-emerging infectious diseases, such as SARS, swine influenza, Middle East respiratory syndrome (MERS) and now COVID-19, indicates that infections requiring critical care, and associated case fatalities, are usually, but not exclusively, concentrated in elderly patients, and in patients suffering from specific comorbidities, such as chronic obstructive pulmonary disease, cardiovascular diseases, diabetes, cancer, chronic kidney disease (CKD).[2-5]

Existing evidence further suggests that infectious disease outbreaks might lead to adverse shocks to the supply of healthcare. Healthcare workers are directly exposed to

risk of transmission and may suffer from the physical and psychological impact of sudden surges in workloads that come with high case fatalities.[6 7] Lack of resources available to healthcare providers and suboptimal quality of health services in a community represent a further source of vulnerability, which hamper the supply of critical care, puts patients at higher risk of negative health outcomes, and endanger the safety of health workers.[8 9]

Despite the fact that, in epidemic intelligence, local disease maps have been in use at least since seminal work by Seaman and Pascalis-Ouvière, who in 1796–1797 employed spot maps to trace cases of yellow fever in New York and Philadelphia,[10 11] interest in how community level variables may moderate the outcomes of infectious diseases was limited before COVID-19, with only a handful of studies reconstructing the distribution of health risks and vulnerabilities across communities in a broad range of settings.[9 11–14] Recent studies have started to address this gap.[15]

In this paper we focus on England, a large part of the UK which by 1 June 2020 was the third-worst hit country for per capita COVID-19 deaths.[16] We collect administration information at the level of the Clinical Commissioning Groups (CCGs), the administrative units that provide National Health System (NHS) services in England, and combine 15 vulnerability indicators in a synthetic Index of Vulnerability.[17] We report geographic vulnerabilities across 191 CCGs and match them with COVID-19 death data up to the 23rd week of 2020. We find a positive relationship between our indicator of vulnerability and the COVID-19 related death rate over the local population, with a correlation coefficient of around 10%.

We also find that vulnerability is not randomly distributed across geographies. Socioeconomically deprived areas display higher prevalence of pre-existing health conditions and lower access to healthcare services. This makes deprived communities disproportionately vulnerable to critical infections and case fatalities during an infectious disease outbreak.[18–24] Identifying these vulnerabilities ahead of time may help to shape preparedness and response policies and, during a pandemic, may be key to fairly allocating the stretched resources of a nation's health systems across communities in need.[14 25–27] In the Methods section, we describe the methods used to build our Index of Vulnerability, which we present in the Results section. In the Discussion section, we present our results and conclude.

## METHODS

Using data gathered from NHS Digital and focusing on the geographic level of CCG areas for England (CCGs, n=191) in the financial year 2018/2019, we combine area-level indicators of vulnerability in an Index of Vulnerability to obtain a standardised synthetic vulnerability measure for each CCG in England. To select relevant vulnerability indicators, which we combine in our Index, we queried PubMed for articles containing the terms 'health',

'inequality', 'vulnerability' and 'pandemics', without language or geographic restrictions. Our search returned 105 results: whereas just 62 studies were published from 1993 to 2019, 43 further papers have been published on the COVID-19 pandemic.

These studies identify three macro-categories of factors that drive geographic differences in health vulnerabilities: (1) disparities in susceptibility to a disease; (2) disparities in the likelihood of contracting a disease and (3) disparities in treatment.[28] Factors that drive disparities in susceptibility include the prevalence of certain pre-existing diseases[29–32] and local demographics, such as a community's age structure.[11 29 31 33–37] There is evidence that the elderly have been hit heavily by the COVID-19 pandemic.[5 37 38] Therefore, in building our Index, we consider (1) the number of people above 70 years old per 10 000 residents across CCGs.

We also consider the prevalence of patients suffering from (2) cardiovascular diseases; (3) COPD; (4) cancer; (5) CKD for patients over the age of 18; (6) hypertension; and (7) diabetes (patients aged 17 or older). These are identified by Public Health England, the English health authority, as characterised by higher risks of severe illness from SARS-CoV-2.[39] To track existing conditions, we used Quality and Outcomes Framework prevalences, computed as the number of patients registered in a general practice with a certain health condition at a given time, divided by the total patients registered at the same general practitioner (GP) and time.

Risk factors associated with the likelihood of contracting the disease include the patterns of population density and mobility[29 36 40–42]; whether a community is principally urban or rural.[11 32 43] Thus, we consider (8) an urban to rural indicator among our vulnerability factors. We also include (9) the ratio of residential and nursing homes per 10 000 population aged over 70 years old. Although there is evidence that a good long-term care infrastructure could reduce hospital admissions and mortality,[44] during pandemic a higher density of home cares represents a risk factor due to the frailty of their residents and observed difficulties in stopping within-facility transmission.[29 32 45] During the ongoing pandemic in England, it also has been suggested that freeing up strained hospital capacity by discharging older patients into care homes may have exacerbated the spread of the disease, although the evidence on this is still mixed.[38 46 47] Based on this, in our Index of Vulnerability, we leverage (9) the ratio of residential and nursing homes per 10 000 population aged over 70 years old; (10) an urban to rural indicator.

While variables (1)–(10) measure quantities that affect the 'demand' of healthcare during a pandemic, extant literature also emphasises the relevance of a set of 'supply'-side factors. Among these, resources available to local healthcare systems, such as the number of intensive care beds and the size of the healthcare workforce, are considered to shape the capacity of local systems to absorb surges in demand.[31 33 40 48 49] The overall quality of a local health system also matters, as high quality practices

shield inpatients from the downside risks of infections, and better protect their healthcare workforce.[29]

To build our Index of Vulnerability, we focus on indicators capturing the NHS workforce in each CCG, and on a set of proxies for the quality of healthcare supply. To measure NHS workforce, we measure (11) the number of nurses within general practices, GPs and hospital staff, including non-medical personnel. We consider full-time equivalents (FTEs), which is a more accurate measure of workforce availability compared with raw headcounts, especially in more deprived and rural areas, where GPs frequently work part-time. We weighted each of these parameters by the CCG-area population. However, while GPs and practice nurses are measured directly at the CCG level, NHS hospital staff is measured at NHS Hospital Trust level. We used the provider code, and the postcode to link the NHS Trusts to the appropriate CCG area.

Following previous literature,[50] we measure quality by including in our Index (12) the percentage of emergency hospital admissions occurring within 30 days of the most recent discharge; (13) the ratio of deaths within 30 days of a non-elective hospital procedure in 2014 (latest data point available); (14) the prevalence of unplanned admissions for ambulatory care sensitive conditions; and (15) emergency admissions for acute conditions that should not usually require hospital admission. Measures (14) and (15) capture acute hospitalisations in patients with specific long-term conditions, when primary care and the delivery of appropriate hospital outpatient services could have prevented deterioration and avoid hospitalisation.[51] All the indicators of quality are indirectly standardised rates based on age and sex specific rates in England.

We combine these 15 indicators in a synthetic index of area-level vulnerability (see online supplemental table S1 for descriptive statistics of all indicators used in this study). To build this index, we first dichotomise each variable by comparing it to the mean of the series in the whole of England. For each variable, value 'one' (more risk) was assigned to CCGs with higher than average target diseases, higher than average elderly population; to CCGs in mostly urban areas; with higher than average prevalence of home and nursing care facilities; lower-than-average healthcare personnel FTEs; and lower than average hospital quality. We assigned value 'zero' for all indicators below the mean.

Building on Organisation for Economic Cooperation and Development guidelines,[52] we obtained a synthetic Index of Vulnerability as the arithmetic sum of these dummies for each CCG area (min=0; max=15, mean=7.38). Therefore, our index represents a functional transformation that, for each CCG area, maps the vector of dichotomous indicators into an integer number. Above-or-below mean indices are a class of additive aggregation methods considered to be robust to outliers. However, they suffer from substantial information loss on the relative distance of each CCG from the national average. We address this issue by presenting a quartile-based indicator in online supplemental file, which offers similar results to the above-or-below average index and thus provide support for our chosen quantification approach.

## Patient and public involvement
Patients and the public were not involved in this research. We would expect that the information provided here can help encourage a public debate when considering what other variables may deserve inclusion in similar indices in the future.

# RESULTS
## Spatial analysis of vulnerabilities in England's CCGs
In online supplemental figures S1–S4, we plot maps for each of the 15 indicators described in the Methods section earlier. We show that, in England, residential and nursing facilities are concentrated in the North and the South West of England, areas also more populated by the elderly. However, looking at the ratio of homecare facilities over the older population, we observe that areas in the North and East Coast have the lowest ratios (online supplemental figure S1). Online supplemental figure S2 depicts the CCG-level distribution of population prevalence for six diseases chosen with the aim of assessing the susceptibility of the system to healthcare demand shocks from the spread of COVID-19.

We find that the areas with the largest prevalence of these underlying diseases, especially cardiovascular diseases, hypertension and CKD, are again located in the North and South West of the country. CCGs in the east of the country, including the East Midlands as well as the Yorkshire and Humber areas are particularly affected by diabetes, COPD and cancer, implying a higher underlying health risk for the population living in these areas, compared with the English average. Increases in the demand for critical care driven by the interaction within infectious diseases and underlying comorbidities raise the question of whether the existing supply of healthcare is adequate, or whether capacity building should be considered.

In online supplemental figure S3, we report on the availability of GPs, nurses in general practices and medical personnel in hospitals. We observe that the healthcare workers are not equally spatially distributed across different areas in England. GPs are more concentrated in the North, the Midlands, and the South West. GP nurses are spread more evenly across the country, and levels of hospital staff are higher in the South and South East, with the lowest values observed in the coastal areas of the East Midlands and the North East of England.

Finally, in online supplemental figure S4, we plot four indicators of hospital quality. We show that there is high variation in these quality markers across England. The North East and North West region of the country (North Kirklees and South Sefton CCGs, for example), report a high number of unplanned hospitalisations, acute emergency admissions and a higher rate of deaths after discharge. High mortality rates appear to be spread more

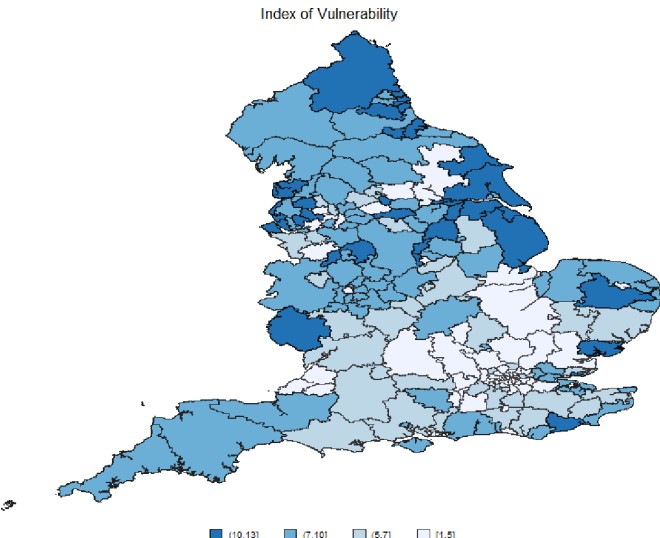

Index of Vulnerability

[10,13] [7,10] [5,7] [1,5]

**Figure 1** Synthetic indicator of vulnerability at Clinical Commissioning Groups (CCGs) level in England. Source: Own elaboration based on National Health System digital data at CCGs areas 2019.

widely, with high levels not only in the North but also in the South (eg, Sussex and Essex CCGs), and especially in the areas of the South West (Devon CCG).

### Synthetic Index of Vulnerability for England

In figure 1, we map the synthetic Index of Vulnerability, obtained with the procedure described in the Methods section. According to this Index, CCGs in the North and the South of England appear most vulnerable to an infectious disease outbreak, with Sunderland, East and North Lincolnshire East, and Wigan Borough among the most vulnerable. Table 1 reports the list of all CCGs by vulnerability score, divided in quartiles. In the first quartile (Q1), we find CCG areas with lower Index of Vulnerability and in Q4 the CCGs area with the highest values. These last group of CCGs should be targeted by the NHS to improve their outcomes and increase the measures of protection in case of pandemic. Online supplemental figure S5 maps the inter-quartile Index of Vulnerability at CCG level, with comparable findings.

To test whether CCGs with similar vulnerability levels are more spatially clustered than expected by chance, we computed Moran's I statistic, one of the most widely adopted tests of spatial correlation between observations. Moran's I can be seen as a correlation index that compares statistical units by weighting each pair by a distance function specified by the means of a spatial weight matrix.[53] We have adopted the contiguity criterium to build the spatial weights matrix and standardised it so that the sum of the elements for each row is unity. Results show that Moran's I, calculated on our vulnerability index, is significant, indicating positive spatial correlation (I=0.155, with a p value of 0.00). This implies that geographically contiguous CCGs tend to show similarly high levels of vulnerability.

### Index of Vulnerability and deaths to COVID-19

Although this paper was conceived in March 2020, before that the pandemic hit hard in England, we check if the vulnerability index could predict the mortality rate due to COVID-19 across areas up to June 2020. We use the deaths registered data in the first 22 weeks of 2020 across CCG areas from Official National Statistics, where the numbers of deaths for COVID-19 are reported. In figure 2, we can observe a positive correlation between mortality rate due to COVID-19 and the Index of Vulnerability. However, some caution needs to be taken when we interpret this result because the data present some limitations. For example, not all deaths have been tested for COVID-19, and this is particularly true for those that happen at home. Most positive tests are recorded in hospitals and this means they are often not reported for the patient's area of residence but rather the CCG in which the hospital is located. These limitations could affect the correlation with the Index of Vulnerability as it is not possible to know accurately the residence of patients with COVID-19.

### DISCUSSION

In this article, we have presented an Index of Vulnerability to map the vulnerability of the English healthcare system to the unexpected and combined consequences of demand-related and supply-related pressures associated with infectious disease outbreaks such as the COVID-19 pandemic. To cope with the pandemic, public health authorities across the country have engaged in significant capacity-building efforts, including opening three temporary hospitals that thankfully were not required, but this demonstrates the type of prompt and decisive action that policymakers should be prepared to pursue when faced with pandemics. In order to maximise the effectiveness of these actions, policy makers need evidence such as that presented in this paper so that limited resources are provided to the areas where they are needed the most.

Evidence from previous infectious disease outbreaks, such as malaria, warns that funding during emergencies tend to be provided in line with health and economic need, but biassed towards richer areas.[54] In light of the correlation between CCG-level vulnerability and deprivation reported in this paper, the distribution of resources in times of emergencies should be guided by community-level factors. Guidelines for 'fair allocation of resources' should be developed based on direct knowledge about local communities—not only considering the health impact of such choices, but also long-run economic and social impacts for the community.

Communities and local healthcare providers should also be involved in these decisions, because community-level knowledge will aid in choosing the right guiding principle—for example, knowledge of population demographics and illness prevalence within a locality might help when deciding whether to maximise benefits/prognosis; to give priority to the worst off; or to reward value to the community. Accurate data collection and dynamic

| Quartile | CCGs |
|---|---|
| **Table 1** | CCGs by quantile Index of Vulnerability |
| Q1 | Barnet; Bath and North East Somerset; Bexley; Brent; Brighton and Hove; Bristol, North Somerset and South Gloucestershire; Bromley; Buckinghamshire; Cambridgeshire and Peterborough; Camden; Castle Point and Rochford; Central London (Westminster); City and Hackney; Crawley; Croydon; Dartford, Gravesham and Swanley; Ealing; East Berkshire; East and North Hertfordshire; Fareham and Gosport; Greenwich; Guildford and Waverley; Hammersmith and Fulham; Haringey; Herts Valleys; Hillingdon; Horsham and Mid Sussex; Hounslow; Islington; Kingston; Lambeth; Leeds; Lewisham; Merton; Mid Essex; Milton Keynes; North East Hampshire; Norwich; Nottingham City; Oxfordshire; Portsmouth; Redbridge; Richmond; South Lincolnshire; Surrey Heath; Tower Hamlets; Vale of York; Waltham Forest; Wandsworth; Warrington; West Cheshire; West Essex; West Kent; West London. |
| Q2 | Ashford; Barking and Dagenham; Bedfordshire; Berkshire West; Bolton; Bradford District; Bury; Calderdale; Canterbury and Coastal; Dorset; East Leicestershire and Rutland; East Surrey; Eastern Cheshire; Enfield; Gloucestershire; Greater Huddersfield; Hambleton, Richmondshire and Whitby; Harrogate and Rural District; Harrow; Havering; High Weald Lewes Havens; Ipswich and East Suffolk; Leicester City; Lincolnshire West; Manchester; Nene; Newark and Sherwood; Newham; North Hampshire; North Norfolk; North West Surrey; Nottingham North and East; Redditch and Bromsgrove; Rushcliffe; Salford; Sheffield; Shropshire; South Cheshire; South Eastern Hampshire; South Warwickshire; South Worcestershire; Southwark; Stafford and Surrounds; Surrey Downs; Sutton; Swindon; Telford and Wrekin; Vale Royal; West Hampshire; West Suffolk; Wiltshire; Wolverhampton. |
| Q3 | Airedale, Wharfedale and Craven; Basildon and Brentwood; Birmingham and Solihull; Blackburn with Darwen; Bradford City; Cannock Chase; Coastal West Sussex; Corby; Coventry and Rugby; Derby and Derbyshire; Devon; Durham Dales, Easington and Sedgefield; East Riding of Yorkshire; East Staffordshire; Eastbourne, Hailsham and Seaford; Fylde and Wyre; Greater Preston; Halton; Hartlepool and Stockton-on-Tees; Hastings and Rother; Herefordshire; Heywood, Middleton and Rochdale; Hull; Isle of Wight; Kernow; Knowsley; Luton; Medway; Morecambe Bay; North Staffordshire; North Tyneside; Northumberland; Nottingham West; Oldham; Rotherham; Sandwell and West Birmingham; Somerset; South East Staffordshire and Seisdon Peninsula; South Kent Coast; South Norfolk; South Tees; South West Lincolnshire; Southampton; Southport and Formby; Stockport; Stoke on Trent; Swale; Thurrock; Trafford; Walsall; Warwickshire North; West Lancashire; West Leicestershire; West Norfolk; Wyre Forest. |
| Q4 | Barnsley; Bassetlaw; Blackpool; Chorley and South Ribble; Darlington; Doncaster; Dudley; East Lancashire; Great Yarmouth and Waveney; Lincolnshire East; Liverpool; Mansfield and Ashfield; Newcastle Gateshead; North Cumbria; North Durham; North East Essex; North East Lincolnshire; North Kirklees; North Lincolnshire; Scarborough and Ryedale; South Sefton; South Tyneside; Southend; St Helens; Sunderland; Tameside and Glossop; Thanet; Wakefield; Wigan Borough; Wirral. |

Source: Own elaboration based on National Health System digital data at CCGs areas 2019.

CCG, Clinical Commissioning Group.

information sharing mechanisms between CCGs—not only of diagnosed cases, but also of staff, medical supplies and excess capacity—will help spread the disease burden and prevent the entire system from collapsing.

No matter what level of community spread has been reached, CCGs should be prepared to act at least in accordance with WHO guidelines.[55] As a minimum, this includes a coordinated strategy across hospitals and communities, and between hospitals and healthcare staff within the community. Planning and stocking medical equipment and drug supplies that are necessary to treat not only COVID-19 patients, but also those with other predisposing illnesses at the local community level is key. Consulting our indicator would allow policymakers and healthcare managers to understand which areas might require strengthened surveillance, monitoring and capacity building so as to minimise the negative outcomes of unexpected health shocks.

Attention should however be paid to the limitations of our Index. The main limitation stems from the selection of indicators of vulnerability. Although our choice was driven by literature, idiosyncratic choices and partial knowledge open the possibility of omitting relevant variables. For example, unlike some existing research,[34] we have not included the availability of beds as a factor that generates supply-side pressures. The reason for this choice is that responses to the COVID-19 crisis suggest that the elasticity in the supply of beds is higher than that of the supply of workforce. In fact, while both bed and healthcare personnel can be increased during a crisis, solutions that could feasibly increase the latter (such as speeding up the promotion of trainees or recalling retirees) have suboptimal implications for quality and organisational efficiency. Moreover, healthcare workers are more exposed to infections, which could create staff shortages, although the hard work of practitioners and managers avoided this to happen in England during the pandemic.

Second, an increasing body of evidence reports a higher risk of infection with COVID-19 and higher mortality in

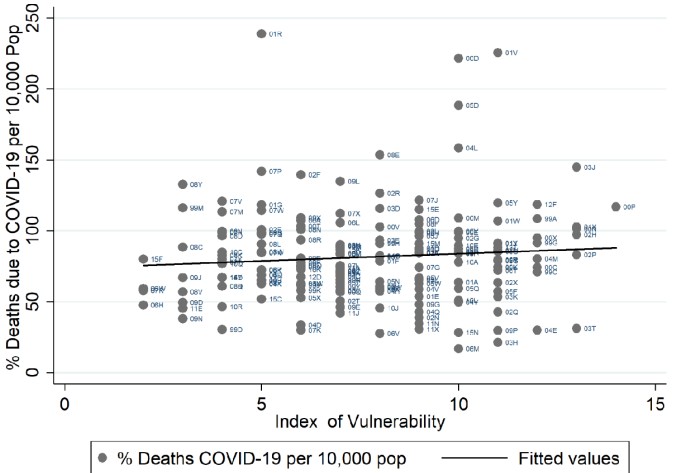

**Figure 2** Correlation between indicator of vulnerability and deaths due to COVID-19 at Clinical Commissioning Group (CCG) areas. Source: Own elaboration based on National Health System digital data at CCGs areas 2019 and the deaths registered data until 22 weeks provided by Official National Statistics 2020.

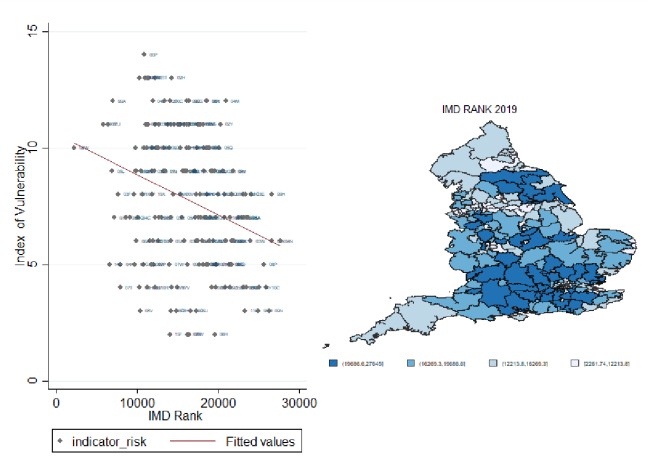

**Figure 3** Correlation between indicator of vulnerability and Index of Multiple Deprivation (IMD) rank. Source: Own elaboration based on National Health System digital data at Clinical Commissioning Groups areas 2019.

individuals belonging to certain disadvantaged ethnic groups.[19 23 24 56–59] However, rather than revealing ethnic and racial disparities, these differences could reflect underlying socioeconomic inequalities, in some cases higher prevalences of pre-existing and at risk health conditions. Other indicators, such as indices of deprivation, might be promising to investigate intersectional vulnerabilities to COVID-19. Deprivation may in fact interact with the vulnerabilities contained in our Index. In fact, social deprivation is associated with the timing of the onset of multimorbidities and the prevalence of long-term conditions.[60] Deprivation also matters for the higher usage of emergency care relative to elective care, 30% of which is not explained by the different case-mix of conditions suffered by patients from a given area.[61] In addition, recruitment of clinicians to deprived areas presents challenges.[62]

In figure 3, we plot our Index of Vulnerability against the Index of Multiple Deprivation (IMD) rank.[63] We show a positive correlation between the two indices. For example, the North East of England shows high levels of both vulnerability and deprivation. Calculating Moran's I local estimation local indicator of spatial association (LISA)[64] for both indices we detect a positive, although weak (about 20%), statistically significant correlation between the two local spatial indices. This shows that there exists a moderate overlap between the clusters detected by the two indices, which we argue indicates that these indices present complementary evidence rather than substitutable evidence. This is more evident when we plot the two indices against the total mortality rate in figure 4. While the Index of Vulnerability shows a high correlation with deaths, the line of best fit between mortality rates and the IMD is quite flat. This is not a surprise as the IMD captures inequality in England well but is heavily influenced by an array of factors (education,

crime, employment, etc), many of which are unlikely to have a major direct role to play in the spread or outcome of a pandemic. Intersectional analysis aimed at evaluating the potentially overlapping role of social deprivation and health factors, such as those employed in our index, is a fruitful area of research as efforts to collect more granular data are mounting. In order to further test the predictive properties of the generated Index of Vulnerability, we analyse the spatial cross-correlation[65] between the amount of deaths in each CCG and both the Index of Vulnerability and IMD, respectively. This allows us to identify how strongly each of these indexes correlates with the amount of death, and thus how well it identifies high risk CCGs, while allowing space to be continuous by using the spatial information as weighting parameters. The results indicate that our proposed Index of Vulnerability performs noticeably better in identifying high risk

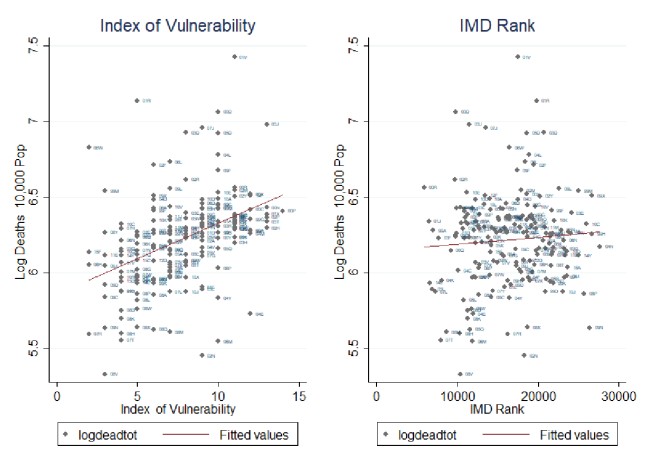

**Figure 4** Correlation of Index of Multiple Deprivation (IMD) rank and Index of Vulnerability versus mortality rate. Source: Own elaboration based on National Health System digital data at Clinical Commissioning Groups areas 2019 and mortality rate of the first 22 weeks of the year 2020.

areas than the IMD, which thus provides support for the proposed index, see online supplemental figure S6.

**Author affiliations**
[1]Nuffield Department of Primary Care Health Sciences, Oxford University, Oxford, UK
[2]CHSEO, University of Oxford, Oxford, UK
[3]Mathematical Institute, Oxford University, Oxford, UK
[4]Oxford Martin School, Unviersity of Oxford, Oxford, UK
[5]Nuffield College, University of Oxford, Oxford, UK
[6]Bocconi Unviersity, Milan, Italy
[7]Institute of Applied Health Research, University of Birmingham, Birmingham, UK
[8]Brunel University of London, London, UK
[9]Department of Economics, Università Ca' Foscari Venezia, Venice, Italy
[10]Hertie School, Berlin, Germany

**Contributors** CN and NC developed the original concept, conducted the analysis and prepared the article draft. SB and FM contributed to the empirical analysis and to drafting the article. DL, MS and SR contributed to the concept of the paper and to writing the article. CN, SR and MS led the data collection. CN, SR and NC led the literature review (research in context).

**Funding** This study was not directly funded. CN and DL are supported by the National Institute for Health Research (NIHR) Applied Research Collaboration (ARC) West Midlands. The views expressed are those of the authors and not necessarily those of the NIHR or Department of Health and Social Care.

**Map disclaimer** The depiction of boundaries on this map does not imply the expression of any opinion whatsoever on the part of *BMJ* (or any member of its group) concerning the legal status of any country, territory, jurisdiction or area or of its authorities. This map is provided without any warranty of any kind, either express or implied.

**Competing interests** None declared.

**Patient and public involvement** Patients and/or the public were not involved in the design, or conduct, or reporting, or dissemination plans of this research.

**Patient consent for publication** Not required.

**Provenance and peer review** Not commissioned; externally peer reviewed.

**Data availability statement** No data are available.

**ORCID iDs**
Catia Nicodemo http://orcid.org/0000-0002-2800-0083
Daniel Lasserson http://orcid.org/0000-0001-8274-5580

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
