## [Reviewer comments · BMJ Open]

ARTICLE DETAILS

TITLE (PROVISIONAL)	Measuring geographical disparities in England at the time of COVID-19: results using a composite indicator of population vulnerability
AUTHORS	Nicodemo, Catia; Barzin, Samira; Lasserson, Daniel; Moscone, F; Redding, Stuart; Shaikh, Mujahed; Cavalli, Nicolo'

VERSION 1 - REVIEW

REVIEWER	Rocco Friebel Department of Health Policy, LSE
REVIEW RETURNED	08-May-2020

GENERAL COMMENTS	Thank you for providing me with the opportunity to review this article for the BMJ Open. The authors of this article present an index of vulnerability by CCG that was constructed using publicly available information on healthcare demand factors and supply-side factors. They argue that this index will be a useful tool for policy makers aiming to direct resource in the fight against pandemics. This research is timely in the context of the current COVID-19 pandemic and could potentially be applied to various other settings (countries) given data availability. However, there are a number of aspects that the authors may want to consider before publication of this article: 1. The authors have used a wide range of demand-side and supply-side variables. The choice of variables seems rather ad-hoc and driven by availability, and it would have been good to see a clear underlying assumption for each indicator selected into the composite measure.2. Have 30-day readmission rates and 30-day mortality rates been risk-adjusted? If not, do they truly reflect on quality of care that allows for comparisons between CCGs? No discussion around the usefulness of these indicators to reflect on quality of care. See here for more information https://bmjopen.bmj.com/content/8/3/e020325.abstract3. In addition to previous point, is it useful to combine readmissions and mortality given that they have been found to be inversely correlated? See here for more information https://onlinelibrary.wiley.com/doi/full/10.1111/1475-6773.127554. The authors argue that ACS admissions reflect on hospital quality of care, but I would assume that they are more linked to primary care access/continuity of care?
---

	5. The index is created by using a large number of indicators (specifically on the demand-side, including underlying health conditions per CCG). Yesterday, the ONS has published information on vulnerabilities by ethnicity, which perhaps should be taken into consideration for this paper. See here for more information https://www.ons.gov.uk/peoplepopulationandcommunity/birthsdeathsandmarriages/deaths/articles/coronavirusrelateddeathsbyethnicgroupenglandandwales/2march2020to10april2020 6. Are there potential mediating factors in the presented relationship? Here, I am thinking about levels of rurality, population density, all of which could reduce the likelihood of infection spreading. 7. The vulnerability index correlates well with the generated synthetic index. What is the value for policy makers to consider the proposed index given that this is only at CCG level, whereas IMD scores are at the LMOA level? 8. It appears that for the current pandemic, COVID mortality does not map onto registered mortality in England. To validate the index, it would be valuable to make an attempt to link both of them, which would strengthen the paper significantly. 9. There are a few consideration/concerns about the creation of the index. a. Equal weighting of indicators: Considering that a large number of demand and supply factors have been selected to create this composite measure, equal weighting without clear justification could lead to biased rankings. At least, I would like to see some discussion around its implication for the index, which is currently not the case. How does the index change when using only demand side factors vs. supply side factors? The picture might look very similar, which would give us confidence that variation is not driven by one particular aspect. b. Have the authors checked for collinearity of components? c. Normalisation and scale adjustment may be required given the selection of indicators. 10. The discussion section is relatively weak and would benefit from further elaboration. Specifically, very little emphasis has been placed on the index itself, though the authors tried to present some of the policy decisions associated with the identification of vulnerable areas. What is the usefulness (strength and limitation) of a composite measure? – see information here: https://qualitysafety.bmj.com/content/28/2/85.abstract Very little discussion has been presented around the choice of indicators. What data would be useful and could further strengthen the appropriateness of the indicator? Finally, the discussion would benefit from a restructuring, i.e., i) discussion of main findings, ii) comparison with previous work, iii) limitations, vi) then policy implication and conclusion. In conclusion, this paper has the potential to make a valuable contribution to the literature and guide policymakers in the future.
--	--

REVIEWER	Sotiris Vadoros King's College London, London, UK; and Harvard T.H. Chan School of Public Health, Boston, USA
REVIEW RETURNED	16-May-2020

GENERAL COMMENTS	This paper creates an index to examine which populations are more vulnerable to unexpected health shocks, in order to inform policy on how to moderate the impact of pandemics. This is clearly a very interesting and topical study of great policy relevance, especially with regards to the current Covid-19 pandemic. It would be useful to mention in the abstract whether you refer to vulnerability in terms of contagious diseases i.e. whether you include factors relating to transmission, or disease in general. It might be that the “respiratory pathogen outbreaks” refer to contagious diseases by definition – but for any non-clinical readers this might not be entirely clear. Similarly, the background section could also include a paragraph on the expected mechanism of what makes a population vulnerable (and how). In the methods section, it is not clear if reference 12 (by Public Health England) is used as a guide for inclusion of all demand-side measures (including people over 70), or only the prevalence of certain diseases (the link to reference 12 by Public Health England does not work, possibly because this content has been moved to another webpage since, so it is not possible to tell which demand-side measures this covers). If it does not include guidance for people over 70, it would be useful to see why this age threshold was used, instead for 65 or 70, for example. The authors acknowledge the limitation of loss of information on the relative distance of each CCG compared to the national average, and therefore introduce a quartile-based indicator, which confirms the results of the main index. However, the results of this quartile-based indicator should also be briefly presented in text (currently they only appear in Figure a1 – unless I am missing something). As the results include Moran’s I statistic, it would be useful to briefly refer to this in the methods section. It would be sufficient to just say for what reason you will be using this method, rather than explaining how it works. The main result of the paper is Table 2 in the Appendix. I would thus suggest moving this to the main text. If there is not enough space, Figures 1-4 in the main text could be moved to the Appendix instead. Apart from the plot in Figure 7, it would also be useful to have numerical value of the correlation. Page 4, last line: “in the Figure 1a in the appendix”: please delete “the” before “Figure 1a”. Page 7: The text refers to Figure 7, which I think should be Figure 6.
--

REVIEWER	Gerald J. Kost, MD, PhD, MS, FAACC POCT•CTR School of Medicine University of California, Davis USA
REVIEW RETURNED	18-May-2020

GENERAL COMMENTS	Straightforward paper with much to offer the reader. The manuscript was submitted April 29th. Time flies when it comes to COVID-19 literature. According to a recent article in Science, "By one estimate, the COVID-19 literature published since January has reached more than 23,000 papers and is doubling every 20 days—among the biggest explosion of scientific literature ever." Therefore, the authors might want to update their text and bibliography just prior to final publication, at least with respect to geospatial concepts and findings. Maybe now there are more than a "handful" of relevant studies. The Conclusion in the Abstract could be more explicit than simply stating "heuristic tool." Abstracts direct search engines and readers to the paper, so a short synopsis of the primary research tool would help that flow of information. The figures appear to be the heart of the paper. It would help if terms were defined in the figure legends, or at least the first time encountered. Likewise, CCG should be spelled out in the Abstract and defined the first time it appears (in the Introduction). In Methods, the authors state, "This indicator allows us to capture the density of older people with high levels of frailty." What about COVID-19 related deaths in this vulnerable group? Why put limitations first in the Discussion? Slows down and distracts the reader. Better last, unless a BMJ Open format requirement. About the statement at the end of the first paragraph in the Discussion, did COVID-19 create staff shortages? It seems to me that the Discussion should start with this statement, "Our results can help to identify vulnerable communities a) that would benefit particularly from early testing; and b) in which isolation of vulnerable population groups would be particularly beneficial." Some reorganization of the text would help the reader to understand main findings. Otherwise, the paper is ready for publication. Thank you for the opportunity of having reviewed it.
--

REVIEWER	Tammy Leonard University of Dallas, United States
REVIEW RETURNED	21-May-2020

GENERAL COMMENTS	Study authors built an indicator of vulnerability for 191 CCGs areas in England by combining information on the prevalence of at-risk individuals within a local community and on the available healthcare supply. Authors state that the indicator may be helpful in targeting locations to strengthen surveillance and monitoring for adverse health shocks. However, study authors do not assess the construct validity of the indicator they've created. Furthermore, they used an adhoc approach to creating the indicator by assuming equal weights and providing no "general equilibrium" stylized assessment regarding the interplay between supply and demand side factors. Above average demand and above average supply are perhaps not a problem, but the indicator, I believe, doesn't contend with these nuances of joint occurrence of potentially off-setting indicators. It's difficult to assess the contribution of this manuscript. The authors claim it can be helpful for targeting additional public health measures, but they don't contend with the issue that the vulnerability indicator itself is in part a function of level of health resources. Is the contribution, thus, to state that where health resources are low and at risk populations are higher, we should increase health resources. This is a very known conclusion and thus not a substantial contribution. Put another way, it's unclear if the vulnerability indicator provides enhanced indication of vulnerability compared to other established or simpler measures. The Moran's I statistic calculated for the vulnerability index was statistically significant, but then the authors stop there and simply assert that there are clusters of high/low vulnerability. Authors should report LISA (local indicators of spatial autocorrelation) statistics. What is the vulnerability index contributing that is unique compared to the established IMD? This could be answered by using LISA's to identify high-high and low-low clusters and assessing the proportion of overlapping clusters identified by the two different indices. Finally, there are other improvements to the vulnerability index that warrant consideration. The index is created by dichotomizing (or in the appendix categorizing) all of the underlying variables. It's unclear why the study authors do not use an approach such as standardizing variables so that more variance (and hence sensitivity) can be retained. The index is created by assigning a value of 1 for supply side factors below average and 0 for above average. Authors also need to state what is assigned when the factor is equal to the average.
---

VERSION 1 – AUTHOR RESPONSE

Reviewer: 1

Reviewer Name: Rocco Friebe

Institution and Country: Department of Health Policy, LSE

Please state any competing interests or state 'None declared': None declared

1. The authors have used a wide range of demand-side and supply-side variables. The choice of variables seems rather ad-hoc and driven by availability, and it would have been good to see a clear underlying assumption for each indicator selected into the composite measure.

We have justified the use of these indicators in the main text. We have specified the literature references that led us to choose the indicators included in our study. We based our choices of age groups and co-morbidity prevalence based on Public Health England reports on at-risk patients and based on peer-reviewed case fatality reports. We have specified that the inclusion of care-homes,

elderly residents, and of the rural-urban indicator should be considered as key factors for accelerated viral transmission. We consider also variables that are critical to delivery of clinical care during the COVID-19 pandemic, namely workforce and care quality (McCabe et al 2020; Vincent & Taccone 2020).. All observational studies of routine databases are limited by availability, but through our use of national datasets, the available data have been previously selected, collected and curated because they are proxies for the critical elements of healthcare services.

2. Have 30-day readmission rates and 30-day mortality rates been risk-adjusted? If not, do they truly reflect on quality of care that allows for comparisons between CCGs? No discussion around the usefulness of these indicators to reflect on quality of care. See here for more information <https://bmjopen.bmj.com/content/8/3/e020325.abstract>

Both indicators are risk-adjusted at gender and age level by NHS Digital. We have added reference to this in the main text.

3. In addition to previous point, is it useful to combine readmissions and mortality given that they have been found to be inversely correlated? See here for more information <https://onlinelibrary.wiley.com/doi/full/10.1111/1475-6773.12755>

We agree with you that high rates of readmissions could end up in high rates of hospital mortality. However, in our analysis we use the indicators separately to capture areas where you have specialised hospitals, such as for example those that focus on cancer, which could have a high rate of mortality and low rate of readmissions, and/or the quality of independent sector providers with high rates of readmissions and low rates of mortality. We have followed the main literature on quality care where both indicators are used (see for more details Doyle et al. 2017).

Doyle Jr, J.J., Graves, J.A. and Gruber, J., 2017. Evaluating measures of hospital quality (No. w23166). National Bureau of Economic Research.

4. The authors argue that ACS admissions reflect on hospital quality of care, but I would assume that they are more linked to primary care access/continuity of care?

We have clarified this point in the text. The ACS indicator reflects not only the primary care quality but also hospital care quality, and in particular the adequate provision of healthcare as a hospital outpatient as per the following definition. See the NHS definition here:

<https://digital.nhs.uk/data-and-information/publications/statistical/nhs-outcomes-framework/may-2020/domain-3-helping-people-to-recover-from-episodes-of-ill-health-or-following-injury-nof/3a-emergency-admissions-for-acute-conditions-that-should-not-usually-require-hospital-admission>.

5. The index is created by using a large number of indicators (specifically on the demand-side, including underlying health conditions per CCG). Yesterday, the ONS has published information on vulnerabilities by ethnicity, which perhaps should be taken into consideration for this paper. See here for more information

We agree that COVID-19 is correlated with ethnicity, although the mechanisms driving this relationship are currently uncertain. While some research that studies this relationship adjusts for age, sex and geography, they do not account for other social and economic factors such as certain types of occupations that ethnic minorities engage in disproportionately to the wider population that may lead to BAME residents being more susceptible to COVID-19. Moreover, certain health conditions that increase the risk of dying such as diabetes, hypertension, and some other chronic conditions, are also more common in some minorities (Kirby 2020). For these reasons, and also because the paper is

focused on vulnerability of areas in case of respiratory diseases and not only COVID, we have elected not to include ethnicity per se, although we do include potential mediators of the relationship such as co-morbidity and deprivation that correlate with ethnicity. Nevertheless, in an attempt to address the reviewer's comment, we approached the Office of National Statistics to collect data on ethnic compositions at the CCG level. This data is collected only once in ten years and is unavailable at the CCG level (the latest available data at a more aggregate level is from 2011).

6. Are there potential mediating factors in the presented relationship? Here, I am thinking about levels of rurality, population density, all of which could reduce the likelihood of infection spreading.

We have considered the rural classification of CCGs as another factor and the analysis now presented includes this variable.

7. The vulnerability index correlates well with the generated synthetic index. What is the value for policy makers to consider the proposed index given that this is only at CCG level, whereas IMD scores are at the LMOA level?

We explain better in the discussion section the relation between the index of vulnerability and the IMD. Using the local spatial correlation test (LISA) we can see that although the correlation is positive, the two indices do not map perfectly and are complementary rather than substitutes. We have constructed the index at a high geographical level because we want to capture the effect of health care services that are usually organised at CCG level and not at LSOA level.

8. It appears that for the current pandemic, COVID mortality does not map onto registered mortality in England. To validate the index, it would be valuable to make an cboth of them, which would strengthen the paper significantly.

We have now considered the correlation with the index and the deaths reported by ONS due to COVID-19. The analysis is reported in the Results section

9. There are a few consideration/concerns about the creation of the index.

a. Equal weighting of indicators: Considering that a large number of demand and supply factors have been selected to create this composite measure, equal weighting without clear justification could lead to biased rankings. At least, I would like to see some discussion around its implication for the index, which is currently not the case. How does the index change when using only demand side factors vs. supply side factors? The picture might look very similar, which would give us confidence that variation is not driven by one particular aspect.

As you will note from other responses, we have explained more clearly why we choose equal weightings and the methods we use to create our index. Furthermore, we have clarified why we have included our chosen indicators and also soften the distinction between demand and supply factors in line with comments of Reviewer 4. Given this softening, we prefer to not create indices based on subsets of our variables because we are mostly concerned on the risk faced in local communities and not the mechanisms by which they may propagate through the system.

b. Have the authors checked for collinearity of components?

We agree that there is some correlation between series included in our indicator and for this reason we have avoided estimations. The main purpose is to explore geographical areas which have a high amount of risk factors. It is very likely that some of these risk factors will be collinear.

c. Normalisation and scale adjustment may be required given the selection of indicators.

The indicators are normalized in line with the methods suggested in the OECD Handbook on Constructing Composite Indicators: Methodology and User Guide (2008). While there are different ways to normalize variables, we use a parsimonious approach by using the mean as the threshold as described in the methods section. This approach is robust against outliers which could have an impact on the synthetic indicator.

10. The discussion section is relatively weak and would benefit from further elaboration. Specifically, very little emphasis has been placed on the index itself, though the authors tried to present some of the policy decisions associated with the identification of vulnerable areas. What is the usefulness (strength and limitation) of a composite measure? – see information here:

<https://qualitysafety.bmj.com/content/28/2/85.abstract>

Very little discussion has been presented around the choice of indicators. What data would be useful and could further strengthen the appropriateness of the indicator? Finally, the discussion would benefit from a restructuring, i.e., i) discussion of main findings, ii) comparison with previous work, iii) limitations, vi) then policy implication and conclusion.

We have rewritten the discussion section in line with these suggestions and the comments of the other reviewers.

Reviewer: 2

Reviewer Name: Sotiris VANDOROS

Institution and Country: King's College London, London, UK; and Harvard T.H. Chan School of Public Health, Boston, USA

Please state any competing interests or state 'None declared': None

It would be useful to mention in the abstract whether you refer to vulnerability in terms of contagious diseases i.e. whether you include factors relating to transmission, or disease in general. It might be that the “respiratory pathogen outbreaks” refer to contagious diseases by definition – but for any non-clinical readers this might not be entirely clear.

The abstract has been edited and now uses the phrase “contagious diseases” instead of “respiratory pathogen outbreaks”.

Similarly, the background section could also include a paragraph on the expected mechanism of what makes a population vulnerable (and how).

We have added text and references explaining how demographic features of local areas, and how resource issues, can affect the outcomes of residents in different local areas.

In the methods section, it is not clear if reference 12 (by Public Health England) is used as a guide for inclusion of all demand-side measures (including people over 70), or only the prevalence of certain diseases (the link to reference 12 by Public Health England does not work, possibly because this content has been moved to another webpage since, so it is not possible to tell which demand-side measures this covers). If it does not include guidance for people over 70, it would be useful to see why this age threshold was used, instead for 65 or 70, for example.

The text has been adjusted to clarify that the guidance was also the reason for focusing on over 70s.

The authors acknowledge the limitation of loss of information on the relative distance of each CCG compared to the national average, and therefore introduce a quartile-based indicator, which confirms the results of the main index. However, the results of this quartile-based indicator should also be briefly presented in text (currently they only appear in Figure a1 – unless I am missing something).

We have moved Table 2 in the text and explained better the results of the inter-quartile index of vulnerability in the main text.

As the results include Moran's I statistic, it would be useful to briefly refer to this in the methods section. It would be sufficient to just say for what reason you will be using this method, rather than explaining how it works.

A paragraph has been added to the "Methods" section to address this.

The main result of the paper is Table 2 in the Appendix. I would thus suggest moving this to the main text. If there is not enough space, Figures 1-4 in the main text could be moved to the Appendix instead.

We have moved Table 2 into the main text and the Figures 1-4 into an Appendix.

Apart from the plot in Figure 7, it would also be useful to have numerical value of the correlation. This information has been added.

Page 4, last line: "in the Figure 1a in the appendix": please delete "the" before "Figure 1a". This has been changed.

Page 7: The text refers to Figure 7, which I think should be Figure 6. We have change the order and the number of the Figures.

Reviewer: 3

Reviewer Name: Gerald J. Kost, MD, PhD, MS, FAACC

Institution and Country: POCT•CTR School of Medicine, University of California, Davis, USA

Please state any competing interests or state 'None declared': None declared.

Time flies when it comes to COVID-19 literature. According to a recent article in Science, "By one estimate, the COVID-19 literature published since January has reached more than 23,000 papers and is doubling every 20 days—among the biggest explosion of scientific literature ever."

Therefore, the authors might want to update their text and bibliography just prior to final publication, at least with respect to geospatial concepts and findings. Maybe now there are more than a "handful" of relevant studies.

We have added a paragraph looking at recently published work on spatial analysis of COVID-19 and attempts to create "risk"-indices.

The Conclusion in the Abstract could be more explicit than simply stating "heuristic tool." Abstracts direct search engines and readers to the paper, so a short synopsis of the primary research tool would help that flow of information.

This has been changed to add further details of the methods used.

The figures appear to be the heart of the paper. It would help if terms were defined in the figure legends, or at least the first time encountered. Likewise, CCG should be spelled out in the Abstract and defined the first time it appears (in the Introduction).

We have rearranged the Figures in the text and improved the legend.

In Methods, the authors state, "This indicator allows us to capture the density of older people with high levels of frailty." What about COVID-19 related deaths in this vulnerable group?

We have added a new section where we look at the correlation with the index of vulnerability and the deaths due to COVID-19. Unfortunately, we do not have the deaths by age at local area levels.

Why put limitations first in the Discussion? Slows down and distracts the reader. Better last, unless a BMJ Open format requirement.

We have re-drafted the discussion section and no longer begin this section with the "limitations".

About the statement at the end of the first paragraph in the Discussion, did COVID-19 create staff shortages?

The data on staffing was obtained prior to the COVID-19 pandemic, and describes the baseline workforce vulnerability. We do not yet have workforce absence data from the hospitals or primary care centres to ascertain how COVID-19 absence from work (due to shielding, sickness, or need to self-isolate due to illness in family members) affected healthcare delivery. However, it would be sensible to assume that areas with a low FTE workforce are more vulnerable to staff shortages.

It seems to me that the Discussion should start with this statement, "Our results can help to identify vulnerable communities a) that would benefit particularly from early testing; and b) in which isolation of vulnerable population groups would be particularly beneficial."

Some reorganization of the text would help the reader to understand main findings. Otherwise, the paper is ready for publication.

The discussion section has been reordered accordingly.

Reviewer: 4

Reviewer Name: Tammy Leonard

Institution and Country: University of Dallas, United States

Please state any competing interests or state 'None declared': None declared

However, study authors do not assess the construct validity of the indicator they've created. Furthermore, they used an adhoc approach to creating the indicator by assuming equal weights and providing no "general equilibrium" stylized assessment regarding the interplay between supply and demand side factors. Above average demand and above average supply are perhaps not a problem, but the indicator, I believe, doesn't contend with these nuances of joint occurrence of potentially off-setting indicators.

We agree that a theoretical framework where the demand and supply equations are used to find the equilibrium would be a very good model. In light with this comment we have softened the distinction between supply and demand as we think that this distinction could distract readers from the key purpose of the paper, which is a primarily a more descriptive analysis. Furthermore, as many of the variables used are correlated we would face serious issues if attempting to perform a more complex estimation.

It's difficult to assess the contribution of this manuscript. The authors claim it can be helpful for targeting additional public health measures, but they don't contend with the issue that the vulnerability indicator itself is in part a function of level of health resources. Is the contribution, thus, to state that where health resources are low and at risk populations are higher, we should increase health resources. This is a very known conclusion and thus not a substantial contribution. Put another way, it's unclear if the vulnerability indicator provides enhanced indication of vulnerability compared to

other established or simpler measures. The Moran's I statistic calculated for the vulnerability index was statistically significant, but then the authors stop there and simply assert that there are clusters of high/low vulnerability. Authors should report LISA (local indicators of spatial autocorrelation) statistics. What is the vulnerability index contributing that is unique compared to the established IMD? This could be answered by using LISA's to identify high-high and low-low clusters and assessing the proportion of overlapping clusters identified by the two different indices.

This is an excellent point, and we have calculated the LISA for both our proposed index and the IMD considering your comment. As we have mentioned in the text, we find a positive, though weak, correlation between the two local spatial indices and they do not map perfectly. This indicates that the two indices are complements more than substitutes. We have added this observation in the discussion.

About the novelty of the contribution, we have presented the correlation between the index of vulnerability and the mortality rate due to COVID-19. We find a positive correlation, although the "deaths registered" data have limitations. Our index shows areas that are more vulnerable to shocks from respiratory pathogens by combining data sources that have previously not been analysed jointly. This gives new insights into methods to plan health and care services, which goes beyond the allocation of resources.

Finally, there are other improvements to the vulnerability index that warrant consideration. The index is created by dichotomizing (or in the appendix categorizing) all of the underlying variables. It's unclear why the study authors do not use an approach such as standardizing variables so that more variance (and hence sensitivity) can be retained. The index is created by assigning a value of 1 for supply side factors below average and 0 for above average. Authors also need to state what is assigned when the factor is equal to the average.

This concern is similar to point 9 raised by Reviewer 1. As we explain in our response to that point, we have clarified why we have included the indicators. The methods employed are suggested in the OECD Handbook on Constructing Composite Indicators: Methodology and User Guide (2008). While there are different ways to normalize variables, we use a parsimonious approach by using the mean as the threshold as described in the methods section. This approach is robust against outliers which could have an impact on the synthetic indicator. Other methods to consider the weight of each indicator are possible, for example principal component analysis.. However, given the high collinearity between these indicators calculating another measure of weights would present significant issues.

Due to the large numbers involved, none of the variables have any observations that are exactly equal to the sample mean. This has been noted in the text.

VERSION 2 – REVIEW

REVIEWER	Rocco Friebel Department of Health Policy, LSE
REVIEW RETURNED	25-Jun-2020

GENERAL COMMENTS	The authors have sufficiently addressed all my previous comments.
---

REVIEWER	Sotiris Vadoros King's College London, UK
REVIEW RETURNED	10-Jul-2020

GENERAL COMMENTS	The authors have addressed my comment
---------------------------------------

REVIEWER	Tammy Leonard University of Dallas, US
REVIEW RETURNED	26-Jun-2020

GENERAL COMMENTS	The paper still does not answer if the index created is any better than the IMD, which is much simpler to compute and already well established. What does it really mean that you detect "about 20%" statistically significant correlation? What new clusters are revealed with your index that are not revealed by IMD? Are these new clusters important because they have high deaths? Your new index is not even correlated with deaths, so I'm uncertain---could it actually perform worse than IMD? Also, this VERY important consideration is tucked into a brief last paragraph of the manuscript. This should be carefully considered as a significant part of your results section.
--

VERSION 2 – AUTHOR RESPONSE

Reviewer: 4

Reviewer Name: Tammy Leonard

Institution and Country: University of Dallas, US

Please state any competing interests or state 'None declared': None

The paper still does not answer if the index created is any better than the IMD, which is much simpler to compute and already well established. What does it really mean that you detect "about 20%" statistically significant correlation? What new clusters are revealed with your index that are not revealed by IMD? Are these new clusters important because they have high deaths? Your new index is not even correlated with deaths, so I'm uncertain---could it actually perform worse than IMD? Also, this VERY important consideration is tucked into a brief last paragraph of the manuscript. This should be carefully considered as a significant part of your results section.

We would like to thank you for these comments. We did not report the correlation between the index of vulnerability and the total mortality as we did not believe that the scope of the paper is to check whether this index of vulnerability is better than the IMD at predicting mortality. The IMD is an index based on seven domains (crime, education, income, employment, disability, housing barriers and living environment) and it does not take into account information relating to the quality of the hospitals, age of the population, or care home cares – all factors likely to affect the consequences of a pandemic – in an area.

However, we have reported in the supplementary file the correlation between the index of vulnerability and IMD correlation versus deaths and can see that the line of best fit between IMD and mortality is flat whereas our index is a good predictor of mortality.

We have additionally tested both the predictive strength of the IMD and Index of Vulnerability for risk by analysing the spatial cross-correlation between each of the aforementioned and mortality rates of

CCGs. This calculation allows us to add further to the standard (Pearson) correlation (Figure S6) and Moran's I discussed in the manuscript's main text. While the Pearson correlation ignores that spatial dimension of the data, Moran's I incorporates the spatial component but solely provides insight into the intra-variable correlation, spatial cross-correlation allows to further test the inter-variable correlation between both indexes and death while allowing space to be continuous and employing it explicitly as a weighting factor. Spatial cross-correlation should thus be interpreted as providing complimentary insight to both of the aforementioned correlation parameters. The results of the analysis are provided in Figure S7. The results indicate that the Index of Vulnerability has significantly better predictive capabilities, identifying approximately 77% of high and low risk areas correctly (as measured by mortality), while when using the IMD only 36% of the CCGs are classified correctly. This supports the conclusion drawn from the Pearson correlation of Figure S6 and provides further strengthens the argument for the proposed Index of Vulnerability as a suitable instrument in identifying high risk areas.

Figure S6: Correlation of IMD Rank and Index of vulnerability versus mortality rate

Source: Own elaboration based on NHS digital data at CCGs areas 2019 and mortality rate of the first 22 weeks of the year 2020.

Figure S7: Spatial Cross-Correlation of Index of Vulnerability (left) and IMD (right) with mortality

Source: Own elaboration based on NHS digital data at CCGs areas 2019 and mortality rate of the first 22 weeks of the year 2020: computed via the SpatialEco R-package

VERSION 3 – REVIEW

REVIEWER	Tammy Leonard University of Dallas, United States
REVIEW RETURNED	04-Aug-2020

GENERAL COMMENTS	Figures S6 and S7 really are helpful to the paper. I'd highly recommend that you include them (or at least highly reference them) in the main paper as these two figures actually are most important for substantiating the contribution of the manuscript. If readers see them, the citations from this paper may increase.
--